# Newly Designed Mesoporous Silica and Organosilica Nanostructures Based on Pentablock Copolymer Templates in Weakly Acidic Media

**DOI:** 10.3390/nano11102522

**Published:** 2021-09-27

**Authors:** Nabanita Pal, Young Sunwoo, Jae-Seo Park, Taeyeon Kim, Eun-Bum Cho

**Affiliations:** 1Department of Physics and Chemistry, Mahatma Gandhi Institute of Technology, Gandipet, Hyderabad 500075, India; nabanitapal_chem@mgit.ac.in; 2Department of Fine Chemistry, Seoul National University of Science and Technology, Seoul 01811, Korea; youngsw02@gmail.com (Y.S.); dedswe2@gmail.com (J.-S.P.); ktyeon91@gmail.com (T.K.)

**Keywords:** mesostructures, pentablock copolymer template, bimodal porosity, boric acid, thermal stability

## Abstract

We developed a new category of porous silica and organosilicas nanostructures in a facile method based on weakly acidic aqueous-ethanol media by utilizing two different pentablock copolymer templates of type PLGA-PEO-PPO-PEO-PLGA. Pluronic block templates were used mainly to prepare these pentablock copolymers with different molecular weights and volume ratios. Silica precursor tetraethyl orthosilicate and organosilicas precursor 1,4-bis(triethoxysilyl)benzene have been used as main source for synthesizing the silica and organosilicas samples. Weak Lewis acids iron(III) chloride hexahydrate, aluminum(III) chloride hexahydrate, and boric acid were utilized as catalyst instead of any strong inorganic acids and the molar ratio of catalyst/precursor has been optimized to 1–2 for preparation of ordered mesostructures. Reaction temperatures have been optimized to 25 °C for pure silica and both 25 °C as well as 40 °C for organosilicas to get the best result for mesostructures. A detailed analysis by using various analytical techniques like synchrotron small angle X-ray scattering, nitrogen sorption, transmission electron microscopy, scanning electron microscope, solid-state ^29^Si CP-MAS nuclear magnetic resonance (NMR), and so on has revealed well developed mesostructures with surface area of 388–836 m^2^/g for silica and 210–691 m^2^/g for organosilica samples, respectively. Furthermore, bimodal typepores have been observed from pore size distribution plot of the samples. Thermal stability of the materials was up to 400 °C as analyzed by thermogravimetric analysis.

## 1. Introduction

Development of a facile method for the synthesis of well-ordered silica and organosilica-based porous materials by utilizing nonionic block copolymer templates in weakly acidic media has become a challenging mission for the scientists all over the world [1,2,3,4,5]. Non-ionic triblock copolymers have already gained enormous attention due to the advantage of generating more stable framework and promoting larger size pores in the silica structure than the conventional surfactants [6,7]. In this connection, pentablock copolymer templates are quite popular nowadays as a non-toxic soft template to generate exceptionally large mesopores of 7–13 nm diameter with different pore morphologies, and this has become a highly demanding research topic for the present generation [8,9]. Moreover, preparation of mesoporous solids with multimodal type adjustable pore structure have achieved much popularity because of the enhanced performances of those in highly promising fields like biomedical research, adsorption, separation study, catalytic applications, and so on [10,11,12]. Recently, our group has demonstrated a bimodal mesoporous silica nanostructured material using CBABC-type pentablock copolymer surfactant [9]. CBABC-type pentablock copolymer refers to a block copolymer consisting of three block chains with a hydrophobic A (PPO) block, a hydrophilic B (PEO) block, and a more hydrophobic C (PLGA) block.

Sometimes, due to the use of strong inorganic acids like HCl, high acidity of the aqueous medium inhibits the proper organization of the self-assembled silica precursors and templates during the sol–gel process, resulting poor mesoporosity in the synthesized material [13]. Therefore, in this study, our aim is to utilize mild acidic catalysts which can be effective in synthesizing ordered mesostructures as well as can minimize the kinetic disorder in the cooperative organization process among self-assembled species caused by strong inorganic acids during the sol–gel reaction [13,14].

Previously, few researchers have proposed various methodologies to prepare large pore silica and organosilicas materials having pore diameter of 7–13 nm by using weak Lewis acids in the solvent media [15,16]. Cho and his co-workers have prepared benzene-silica periodic mesoporous organosilica (PMO) under mild acidic conditions using different metal salts like iron chloride, cobalt and nickel chloride, etc. as catalysts [10]. Aluminum chloride and boric acid also have been successfully utilized for synthesizing benzene and thiophene based periodic mesoporous organosilicas [15,16]. However, a suitable facile route for the preparation of large pore organosilica nanoparticles by using pentablock copolymer surfactants and mild acid catalysts is very few in literature.

Herein, we demonstrate a facile synthesis of mesostructured silica and benzene-silica by employing tetraethyl orthosilicate (TEOS) as a silica precursor and 1,4-bis(triethoxy-silyl)benzene (BTEB) as an organosilicas precursor, respectively, using two kinds of long PLGA-PEO-PPO-PEO-PLGA pentablock copolymer, poly(lactic acid-*co*-glycolic acid)-*b*-poly(ethylene oxide)-*b*-poly(propylene oxide)-*b*-poly(ethylene oxide)-*b*-poly(lactic acid-*co*-glycolic acid) surfactants in ethanol-water media in presence of mild acid catalysts, namely, iron(III) chloride, aluminum(III) chloride, and boric acid. The resulting materials exhibit high surface area having large mesopores along with bimodal pores. Furthermore, we have explored the effect of changing the synthesis temperature, copolymer templates as well as concentration of precursors on the nature of generated porosity, ordering and thermal stability of the synthesized silica and organosilica frameworks. It has been observed that silica and organosilica materials prepared using boric acid as catalyst are more stable and have more ordered mesostructures compared to those obtained using other catalysts. The novelty of this work is to present a facile method to synthesize partially-ordered mesostructured silica and organosilica having bimodal porosity that can be tailored using water-soluble pentablock copolymer templates, especially by adjusting mild acidic conditions and also by changing the synthesis temperature. In addition, this work suggests structural information about micellar behavior of pentablock copolymer templates with hydrophobic block at both end groups in weak acidic aqueous solutions.

## 2. Experimental Section

### 2.1. Materials

Pentablock copolymer templates PLGF108-220 and PLGF108-225 were synthesized in our laboratory using Pluronic block copolymers following the previous procedure [9,17]. The molecular weights and compositions of these two kinds of pentablock copolymers are summarized in Table 1. Tetraethyl orthosilicate (TEOS, Si(OEt)_4_) and 1,4-bis(triethoxysilyl)benzene (BTEB, (EtO)_3_Si-Ph-Si(OEt)_3_) were used as silica and organosilica precursors which were obtained from Sigma-Aldrich (Burlington, MA, USA). Metal salts like iron(III) chloride hexahydrate (FeCl_3_·6H_2_O, Aldrich), aluminium(III) chloride hexahydrate (AlCl_3_·6H_2_O, Aldrich), and boric acid (H_3_BO_3_, Aldrich) were used as acid catalyst. Distilled water prepared in our laboratory was used as a main solvent for sol–gel reactions. Ethanol (EtOH, ~98%, Aldrich) was also used as a co-solvent to control hydrolysis-condensation kinetics and solubility of pentablock copolymers.

### 2.2. Synthesis Procedure

In a typical synthesis of PMS samples, 1.5 g of pentablock copolymer template was dissolved in a mixture of 60 g of distilled water and 10 g of EtOH in a glass bottle followed by magnetic stirring at 25 °C for 2 h. Next, acid catalyst (FeCl_3_, AlCl_3_ or H_3_BO_3_) was added to the above solution with a fixed molar ratio of acid catalyst/TEOS = 2 and the mixture was stirred for another 1 h. After that, 0.8 g of TEOS (for PMSF-1 sample) was added to the mixture followed by continuous stirring for 20 h. The final mixture was kept in a 100 °C temperature convection oven for 24 h under a quiescent condition. Finally, the precipitate mixture was filtered with washing solvents like acetone and ethanol using an aspirator at room temperature. This as-synthesized product was calcined at 550 °C temperature in air for 5 h to remove organic template. Other silica samples listed in Table 2A were prepared following the similar procedure using different amount of the TEOS and acid catalysts were named as PMSF, PMSA, and PMSB based on the acid catalyst FeCl_3_, AlCl_3_, and H_3_BO_3_ used, respectively. Organosilica samples (Table 2B) were also prepared in an analogous method using BTEB and named as PMOF, PMOA, and PMOB, respectively. Organosilica samples synthesized at elevated temperature (i.e., 40 °C) were named as PMOFH, PMOAH, and PMOBH, respectively. A schematic representation for the synthesis procedure is shown in Figure 1.

### 2.3. Characterization

The small-angle X-ray scattering (SAXS) patterns were obtained using synchrotron radiation source (*E* = 10.5199 keV, *λ* = 1.1785 Å) of 3C and 4C beam lines in Pohang Accelerator Laboratory (PAL). The distance between sample and detector was selected as 3 m to obtain the exact Bragg spacing value. Each powder sample was firmly anchored inside a cavity in a multi-sample holder using a Kapton tape. Beam exposure times were in the range 1 to 2 s for each sample.

Nitrogen adsorption-desorption isotherms were obtained on an ASAP 2420 analyser (Micromeritics, Norcross, GA, USA). The samples were degassed at 110 °C below 30 μmHg for around 5 h before adsorption–desorption of nitrogen gas at −196 °C. The Brunauer–Emmet–Teller (BET) specific surface area was evaluated by using adsorption data in the relative pressure (*P*/*P*_0_) range of 0.04 to 0.20. The total pore volume (*V_t_*) was obtained from the amount adsorbed at 0.99 *P*/*P*_0_. The pore size distribution (PSD) was calculated from adsorption isotherms by using the KJS (Kruk–Jaroniec–Sayari) method [18]. The mesopores volume and micropore volume were estimated between 2–50 nm and below 2 nm, respectively, by integration of the PSD curve.

Transmission electron microscope (TEM) images were obtained using a JEM-2010 microscope (JEOL, Tokyo, Japan) operated at 200 kV. The powder samples were placed on carbon-coated copper grid before TEM measurement. Field emission scanning electron microscopy (FE-SEM) images were obtained using a VEGA3 instrument (TESCAN, Seoul, Korea) at 20.0 kV. The powder samples were placed on carbon film followed by Pt coating before SEM measurement.

Trace of metal ions present in the samples were determined by using an inductively coupled plasma optical emission spectrometer (ICP-OES, Jobin Yvon Ultima 2C) (HORIBA, Palaiseau, France) with the wavelength to 395.254 nm at the Korea Basic Science Institute (KBSI) Seoul Centre. Final data were obtained by averaging data measured by three times.

Solid-state ^29^Si CP-MAS NMR spectra were obtained utilizing 400 MHz AVANCE II instrument (Bruker, Ettlingen, Germany) at the Seoul Western Center of Korea Basic Science Institute. Each spectrum was taken from the following experimental conditions: 6 kHz of spinning rate, 3 s of delay time, and 79.488 MHz of radio frequency. The chemical shifts were obtained with reference to the peak of tetramethylsilane (TMS).

Thermogravimetric (TG) analysis was performed using a high-resolution mode of the Q50 analyzer (TA Instrument, New Castle, DE, USA). TG and DTG profiles have been recorded up to 800 °C with a heating rate of 10 °C min^−1^ in flowing 40 mL of air and 60 mL of nitrogen.

## 3. Results

Using weak Lewis acid listed in Table 2, a series of mesoporous silica and organosilica samples have been prepared here in presence of two types of PLGA-PEO-PPO-PEO-PLGA pentablock copolymer templates named PLGF108-220 and PLGF108-225 as listed in Table 1. TEOS and BTEB have been chosen as silica and organosilica precursors, respectively, and no other strong acids or any additive salts are used here. Silica samples prepared using molar ratio of FeCl_3_·6H_2_O/TEOS = 2 did not show any porosity. Silica samples prepared using AlCl_3_·6H_2_O/TEOS = 2 and H_3_BO_3_/TEOS = 2 show ordered porosity, so some additional samples with varying quantities of TEOS have been prepared. In a similar way, organosilica with different molar ratios of acid catalyst/BTEB and variation of temperature, have been prepared. The details of the experimental conditions are summarized in Table 2.

### 3.1. TEM Images

To verify the nanostructure and porosity of the synthesized silica and organosilicas materials, TEM images were recorded and those are presented in Figure 1, Figure 2 and Figure 3. In case of silica samples prepared using AlCl_3_ and H_3_BO_3_ as catalysts (PMSA and PMSB) at room temperature, mostly wormhole-like mesopores of 5–8 nm dimension are observed being distributed throughout the specimen. Fairly ordered porosity with mesopore diameter has been observed in case of silica sample PMSB-3 synthesized using H_3_BO_3_, 0.4 g of TEOS, and PLGF108-220 template. On the other hand, TEM image of silica sample (PMSF) synthesized based on FeCl_3_ catalyst shows nonporous structure (Figure 1).

TEM images of organosilica samples synthesized at room temperature (25 °C) using BTEB are shown in Figure 2. Samples prepared using FeCl_3_ catalyst and both the pentablock copolymer templates PLGF108-220 and PLGF108-225 show wormhole like porosity with pore diameter 3–5 nm (Figure 2(a-1–4)). Other organosilica samples synthesized using AlCl_3_ and H_3_BO_3_ acid catalysts exhibit well-defined wormhole like porosity as well as highly ordered hexagonal (*p*6*mm*) mesostructure [19]. It is observed that, with 1.0 mL organosilica precursor, using both the templates and AlCl_3_ catalysts more ordered mesoporous structure is obtained (Figure 2(b-1–4)), whereas, in case of H_3_BO_3_, BTEB amount 1.8 mL has been optimized to give distinct mesoporosity (Figure 2(c-1–4)). Pore diameter for all these sample are in the range of 4–8 nm.

TEM images for the organosilica samples synthesized at elevated temperature (40 °C) have shown a significant change in the mesostructures (Figure 3a–c). All the samples prepared using AlCl_3_ (PMOAH) show now well-defined wormhole like pore arrangements, while PMOBH organosilicas have more ordered nanostructures with uniform hexagonal mesoporosity. No significant change is observed in the images of PMOFH samples due to temperature effect.

### 3.2. SAXS Patterns

Mesoporosity of the silica and organosilica nanostructures has been further confirmed by synchrotron small-angle X-ray scattering (SAXS) analysis as displayed in Figure 4, Figure 5 and Figure 6. No peak observed in SAXS pattern gives evidence about the non-porosity of silica samples synthesized using FeCl_3_ (Figure 4a). In PMSAs silica, one peak observed in SAXS indicating presence of mesoporosity in those samples. On the other hand, more than two peaks, but not exactly indexed as (100), (110), and (200) planes of ordered hexagonal (*p*6*mm*) mesostructure [20,21], are observed in case of PMSB samples (Figure 4c-A–C), although other PMSB samples (Figure 4c-D–F), do not show much significant ordered porosity. This observation suggests that samples prepared using PLGF108-220 shows better porosity compared to that using PLGF108-225 template. All the organosilica samples show good mesoporosity with multiple peaks, but small and broad peaks do not represent highly ordered nanostructure (Figure 5a–c). The porous nature of the samples is still maintained during high temperature synthesis (Figure 6a–c). Because ordered porosity of most of the organosilica samples is not clearly detected from the SAXS pattern, the aforementioned TEM images (Figure 1, Figure 2 and Figure 3) seems to be a partially ordered mesostructure. Bragg’s spacing (*d*) values for these samples are calculated based on 2π/*q**, where *q** is the *q*-value at the maximum of SAXS peak in all the SAXS patterns and the values ranging from 12.60 to 23.20 nm are shown in Table 3. The values were calculated with the first peak for each sample, however the overall first peaks are very small, representing that the values are not the general largest pore-to-pore distance but the trace attributed from partial and small portion of lager pores. Furthermore, multiple SAXS peaks strongly suggest bimodal or overlapped multimodal nanoporous structure.

### 3.3. FE-SEM Images

The morphology of the synthesized silica and organosilica nanoparticles has been obtained from the FE-SEM image recording data which are shown in Appendix A.

### 3.4. N_2_ Adsorption-Desorption Analysis

Valuable information about BET surface area and pore width are achieved from N_2_ adsorption analysis measurements, along with the corresponding pore size distribute plot [18,21], which is given here in Figure 7a, Figure 8a and Figure 9a.

Respective pore size distribution (PSD) plots for the samples obtained applying improved KJS method [18] to the adsorption isotherms are shown in Figure 7b, Figure 8b, and Figure 9b.

### 3.5. Elemental Analysis

Elemental analysis has been carried out using ICP-AES to find out trace of elements like Fe, Al, or B present in the pure silica and organosilica samples. The data taken for some samples are shown in Table 4. Trace quantities of Al and B species (concentration 650–665 ppm and 20–168 ppm, respectively) have been found to be incorporated in organosilica samples while preparing using AlCl_3_ and H_3_BO_3_. Therefore, it is evident that, here, Lewis acids AlCl_3_ and H_3_BO_3_ are truly acting as catalysts maintaining weakly acidic media to form the mesostructures.

### 3.6. ^29^Si CP-MAS NMR

^29^Si CP-MAS NMR of PMOA and PMOB samples have been recorded and shown in Figure 10 in order to know the successful chemical linkage of covalently bonded mesoporous organosilica samples. Signals for PMOA-2, PMOA-4 observed around −62.0 (−61.9) ppm, −70.6 ppm and −77.9 (−79.4) ppm, whereas for samples PMOB-1, PMOB-3 chemical shift signals are visible near −61.5 (−62.2) ppm, −70.6 ppm, and −79.6 ppm, respectively. These signals clearly suggest chemical linkages of type *T*^1^ (C–Si-(OSi)(OH)_2_), *T*^2^ (C–Si-(OSi)_2_(OH)), and *T*^3^ (C–Si-(OSi)_3_), respectively [13], indicating proper cross-linked network structures in BTEB units located in the organosilica.

### 3.7. TG-DTA Thermogram

In order to get an idea about the stability of benzene-loaded organosilica samples, TG and DTA measurements have been performed and the data are shown in Figure 11.

## 4. Discussion

The results from TEM analysis (Figure 1, Figure 2 and Figure 3) clearly indicate that, for both silica and organosilica samples, H_3_BO_3_ has been proved to be more appropriate acid catalyst to generate prominent mesostructures containing ordered structure. Moreover, for organosilica, elevated temperature (40 °C) synthesis is more suitable although AlCl_3_ fits for room temperature synthesis. Furthermore, this type of temperature effect is not observed in case of organosilica prepared using iron catalyst. Compared to our previous reports related with triblock copolymer templates [13,15,16], it seems that the ordered mesostructure is related with kinetic order for all kinds of reactant species, especially for long-chain pentablock copolymer templates as used in this study. In addition, TEM images suggest that pentablock copolymer chains are not easy to bend to form single micelles with loop-typed corona under weak acidic conditions.

From FE-SEM images (Appendix A), it is evident that silica nanoparticles prepared using boric acid catalyst (PMSB) show uniform spherical shaped particles of 100 nm to 1 μm distributed throughout the specimen, while in some cases particles are found to be agglomerated, as visible in Appendix A. On the contrary, organosilica samples are observed as spherical but non-uniform and larger particle size ranging from 1 to 10 μm. In most of the samples the particles have been agglomerated as shown from Appendix A. Furthermore, synthesis at higher temperature demonstrates a decrease in spherical morphology and loss of uniformity which is observed in Appendix A. This type of temperature effect on the morphology of synthesized nanoparticles has also been observed previously in other works [22,23].

N_2_ adsorption–desorption isotherms plotted for PMSA and PMSB silica samples as well as those for organosilica samples prepared with AlCl_3_ and H_3_BO_3_ are shown in Figure 7, Figure 8 and Figure 9. All the samples exhibit type IV adsorption isotherm which is typical for mesoporous samples (according to IUPAC classification) [24]. A gradual increase of adsorption branches at *P*/*P*_0_ = 0.0–0.4, then a steep rise at *P*/*P*_0_ = 0.4–0.50 with H1 type hysteresis loops in the desorption branches is observed in the silica and organosilica samples which obviously indicate the capillary condensation of nitrogen in cylindrical mesopores [22]. Estimated BET specific surface areas of the samples (*S*_BET_) from the isotherms are mentioned in Table 3. Silica samples PMSA and PMSB show high specific surface area of approximately 700–800 m^2^g^−1^, although a little decrease in surface area has been observed due to change of template from PLGF108-220 to PLGF108-225. In case of benzene functionalized organosilica PMOAs and PMOBs, BET surface area has been significantly reduced to the range of 320 to 690 m^2^g^−1^ as well as the micropore and mesopores volume also decrease notably. An almost similar value of surface area is observed for the samples PMOBHs which suggests retention of mesoporosity during the synthesis at 40 °C temperature (Figure 9a).

Narrow PSD is observed for all organosilica samples indicating considerable uniformity of mesostructures while a bimodal type PSD plot in case of silica samples (Figure 7b) suggests two types of pore diameters is observed in the sample [9,25,26]. The pore diameters measured at the maxima of the PSD plots (*D*_KJS_) vary significantly from 2.8 to 9.5 nm for PMSA and PMSB silica samples, whereas a similar range of pore diameter (5.27–6.57 nm) is shown by organosilica samples. On the other hand, it is evident that samples synthesized at high temperature (PMOAH and PMOBH) show bimodal type PSD due to formation of large mesopores having pore diameter in the range of 8.9 to 13.6 nm (Figure 9). The bimodal pore size distribution is due to the formation of non-uniform micellar-silica complexes, and is considered to be fundamentally due to the following two causes: (i) A penta-block copolymer with a large molecular weight in which the hydrophobic block exists at both end groups does not effectively overcome entropy when forming micelles. (ii) The kinetic disorder (i.e., rate imbalance) in the cooperative self-assembly reaction between silica and polymer template.

A closer analysis of TG-DTA curves (Figure 11) reveals that thermal degradation starts after 400 °C for organosilica sample prepared with boric acid at room temperature (PMOB-1) whereas it is slightly before 400 °C in case of benzene-silica sample (PMOA-2) synthesized using aluminium chloride as catalyst. Synthesis of PMO at elevated temperature (PMOBH-1) that is at 40 °C does not show much difference in the TG pattern of benzene-silica prepared using boric acid. Although, in case of PMOAH-2, there is a little increase in the degradation temperature observed in Figure 11C. Therefore, it is apparent from the above result that thermal stability of benzene silica PMO is increased due to use of boric acid as catalyst for preparation. The thermal stability of samples synthesized at high temperatures is also maintained. Higher stability of organosilica using boric acid catalyst has also been reported previously in other literature work [16,26,27].

Thus, verifying all the results, it is quite obvious that according to ICP-AES data very negligible quantity of boron or aluminum is present in the silica samples. From solid-state ^29^Si CP-MAS NMR spectra, it is also clear that silica Si-O- and –Si-C_6_H_4_-Si bonds are present only in the prepared silica and organosilica samples. Therefore, those Lewis acids have mainly played the role of a catalyst to facilitate in the self-assembly process of silica precursors and PLGF co-polymer template. Moreover, in comparison to aluminum and iron acid catalysts, boric acid has been proved much more effective in generating highly stable ordered mesostructured silica and organosilica materials under ambient as well as higher temperature reaction conditions.

## 5. Conclusions

Instead of using any strong inorganic acid, mild acid catalysts like boric acid, aluminum, and iron chloride have been utilized for the preparation of well-developed mesoporous silica and benzene-silica mesostructures using two kinds of PLGA-PEO-PPO-PPO-PLGA pentablock copolymers as surfactants. Boric acid has been proved more efficient compared to the other Lewis acids for generating mesostructures with bimodal porosity in the synthesis conditions at both ambient and elevated temperature. Thorough investigation has been done to evaluate any template effect due to molecular weight and volume ratio of each block in pentablock copolymer. Silica samples using a variable amount of TEOS precursor and iron(III) chloride catalyst exhibit nonporous samples, while mesoporous structure have been obtained using aluminum(III) chloride hexahydrate and boric acid. Among those, a silica sample prepared with boric acid showed partially-ordered mesostructures with very high surface area up to 836 m^2^/g. Organosilica samples synthesized with BTEB precursor and boric acid, also shows superior result compared to other organosilica synthesized using FeCl_3_ or AlCl_3_. In addition, those samples prepared at 25 °C have higher surface area (i.e., 320~691 m^2^/g) than those (i.e., 210~361 m^2^/g) synthesized at elevated temperature say 40 °C. Furthermore, the pore size distribution (PSD) plot showed narrower unimodal distribution for samples prepared at 25 °C, whereas the samples prepared at 40 °C show bimodal distribution with larger mesopores over 10 nm. TEM images exhibit ordered and partially-ordered mesostructures for all these mesoporous benzene-silica samples. As for the thermal stability, mesoporous organosilica samples prepared with boric acid are thermally more stable than the other samples prepared with AlCl_3_. Therefore, this study demonstrates that mesostructure can be tailored using water-soluble pentablock copolymer templates especially by adjusting mild acidic conditions.

## Data Availability

Not applicable.

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
