# Peer review of "Newly Designed Mesoporous Silica and Organosilica Nanostructures Based on Pentablock Copolymer Templates in Weakly Acidic Media"

_nanomaterials, 2021, doi:10.3390/nano11102522_

Round 1

Reviewer 2 Report

In this study, demonstrated a facile synthesis of mesostructured silica and benzene-silica by employing tetraethyl orthosilicate (TEOS) as a silica precursor and 1,4-bis(triethoxysilyl)benzene (BTEB) as an organosilicas precursor by using PLGA-PEO-PPO-PEO-PLGA pentablock copolymer surfactants in ethanol-water media in presence of mild acid catalysts, iron(III) chloride, aluminum(III) chloride and boric acid. The resulting materials exhibit high surface area having large mesopores along with bimodal pores. It has been observed that, silica and organosilica materials prepared using boric acid as catalyst, are more stable and have more ordered mesostructures compared to those obtained using other catalysts. The results presented in this report are promising with good prospects to synthesize partially-ordered mesostructured silica and organosilica having bimodal porosity. Thus, I recommend it for publication in Nanomaterials after fixing several issues.

Major issues:

  1. Earlier author report similar kind of work (J. Phys. Chem. C 2018, 122, 4507−4516), on that study strong acid and weak acid has been used for mesoporous silica synthesis (Iron(III) chloride/TEOS molar ratio = 5) Why did in this study author fix molar ratio of acid catalyst/TEOS = 2?
  2. Need to scientific short explanation FeCl3 not acting as catalysts to form the mesostructure in this study in comparison with, AlCl3 or H3BO3.

Minor issues:

  1. PLGA-PEO-PPO-PEO-PLGA full name need to be used.
  2. Page 3 Scheme 1. 1,4-bis(triethoxy- 42 silyl)benzene chemical structure need to be fixed.
  3. Page 4 line 5, 6, 33 between word space missing.

Author Response

This manuscript is a resubmission of an earlier submission. The following is a list of the peer review reports and author responses from that submission.

Round 1

Reviewer 1 Report

Newly Designed Mesoporous Silica and Organosilica Nanostructures Based on Pentablock Copolymer Templates in Weakly Acidic Media, Nabanita Pal, Young Sunwoo, Jae-Seo Park, Taeyeon Kim, Eun-Bum Cho

The paper presents a method using mild acidic conditions (contrasting previous strong acidic conditions) to produce porous silica and organosilica based nanoparticles. The samples produced are characterized by a large number of techniques: SAXS, TEM, SEM, nitrogen sorption, solid state NMR, thermogravimetric analysis and spectrometry.

I find the material presented relevant for Nanomaterials and also acknowledge the very thorough collection of data to characterize the samples synthesized. However, I find that the paper is lacking in discussion of results and explanation of observations. Also, analysis of some of the data, especially the SAXS data (see below), is very simple and basic and perhaps even wrong. More information can be obtained from these data.

Some specific points:

  • Give a better introduction to the method of synthesis by using a schematic drawing, emphasizing the role played by the template block copolymer. Give the chemical structure formula – not just names - for all basic ingredients.
  • P2l51: What is meant by ‘kinetic disharmony’? Also used l356.
  • P2 l54: How large are ‘large pores’?
  • Table 1 – why is the number of repeating GA units given as 13 and 17 – according to stoichiometry it should be 14 and 18 (as is found by the other three units)
  • SAXS: How thick were the samples? Could multiple scattering be a problem? What is meant on p3l124: The Sample-detector distance was varied to obtain exact Bragg spacing value? If the data were measured on an absolute scale, it is possible to obtain the inner surface area for a porous sample from the SAXS data and compare with BET values.
  • 1-3 Scale bar should be better visible. Images are very small and details hardly recognizable.
  • SAXS: The headline on p.7 is SAXS images – more correct is SAXS curves. A peak in the SAXS curve is interpreted to mean presence of mesoporosity without any argumentation or model references. The indexing of hexagonal peaks should be documented more specifically. The validity of the calculation of Bragg spacings using 2π/q* depends on the presence of an ordered structure – and also on the nature of the ordering. A full model fit to the data would provide more (and more secure) information. If the order and porosity assumed is not consistent with TEM data – how can the authors be certain that the SAXS contrast arises form porosity and not other density fluctuations? The value of characteristic distances are given in table 3 with 1/10 Å precision – is that realistic?
  • The bimodal porosity in some samples should be explained in terms of structure

The abstract especially will benefit from editing content, language and minor errors.

Reviewer 2 Report

The paper by Pan et al. is devoted to the preparation and investigation of ordered porous mesostructures. The goal has been to find facile synthesis routes to hexagonal mesostructured silica and benzene-silica from tetraethyl orthosilicate and bis(triethoxysilyl)benzene. Mildly acidic iron and aluminum chloride hexahydrates as well as boric acid were used as catalysts. (The absence of residues possibly arising from the catalysts was proven by ICP-AES) Reaction temperatures, catalyst-precursor ratios, effect of templates, precursor concentrations, and post-synthesis thermal conditions were changed in order to investigate the qualities and stability of the prepared structures. The synthesized materials were thoroughly characterized using SAXS, nitrogen sorption studies to determine BET surface areas, TEM, SEM, NMR. The results revealed clear differences between the influences of different catalysts. Iron-based catalyst did not enable formation of porous networks.In comparison to aluminum and iron acid catalysts, boric acid was most effective for the synthesis of chemically and thermally stable ordered mesostructured silicas and organosilicas.

The paper comprises complex studies represented in a very detailed manner. The topic may not appear very easy to follow for a reader. The again, the results, as presented, are quite clear and confirm the goals and hypotheses outlined in the description of the background. It would be hard to recommend any more concise manner of presentation. The paper might be published almost in the present form.

The only issue worth addressing is that the authors should double check the manuscript for linguistic errors. Linguistic errors, typos, or awkward syntax can be met, for instance, in rows nos. 22, 23, 423-424. Another small notice on row 141: full name for the acronym ICP-AES is inductively coupled plasma optical atomic spectrometer.

An additional comment can be made on the following: in the row no. 360, it is said that that „From FESEM images (Fig. S1-3) its evident, that , silica nanoparticles....“ At first, instead of „its“, one should write „it is“. Secondly (just commenting it, not suggesting changes), it is always hard to understand, if/why people sometimes choose putting data or figures, essential to follow along with their manuscript Discussion, into supplementary material.

Round 2

Reviewer 1 Report

nanomaterials-1247974

Newly Designed Mesoporous Silica and Organosilica Nanostructures Based on Pentablock Copolymer Templates in Weakly Acidic Media, Nabanita Pal, Young Sunwoo, Jae-Seo Park, Taeyeon Kim, Eun-Bum Cho

In my first report, I wrote:

‘I find the material presented relevant for Nanomaterials and also acknowledge the very thorough collection of data to characterize the samples synthesized. However, I find that the paper is lacking in discussion of results and explanation of observations. Also, analysis of some of the data, especially the SAXS data (see below), is very simple and basic and perhaps even wrong. More information can be obtained from these data.’

While the authors have addressed the list of more specific points, they generally ignore my main criticism: that the paper is lacking in discussion of results and mechanistic explanations of observations. E.g. in the answer to my previous report, the authors write: “Simply, kinetic disharmony (disorder) means rate imbalance. In fact, (ordered-) mesopores can be prepared three kinds of forces, i.e. i) driving force to form micelles between block copolymer, ii) sol-gel condensation reaction of silica precursors, and iii) interfacial force between PEO block of block copolymer chains and silica sources. If the three forces are not optimized, (ordered-) mesopores can’t be prepared. “ A more in-depth explanation along these lines of the mechanisms behind the observed structural differences using different synthetic pathways would improve the manuscript.

The authors do add in the discussion: ‘The bimodal pore size distribution is due to the formation of non-uniform micellar-silica complexes, and is considered to be fundamentally due to the following two causes. i) A penta-block copolymer with a large molecular weight in which the hydrophobic block exists at both end groups does not effectively overcome entropy when forming micelles, ii) The kinetic disorder (i.e. rate imbalance) in the cooperative self-assembly reaction between silica and polymer template.' However, I simply do not understand the authors arguments. How does the argumentation relate to scheme 1?

The addition of scheme 1 to the paper is very useful.

Regarding SAXS:

The authors still don’t give the sample thickness. Multiple scattering might be a problem – see e.g.  Brian Richard Pauw 2013 J. Phys.: Condens. Matter 25 383201.

The authors still don’t document that the presence in the SAXS curve of a peak indicates porosity (and not other electron density fluctuations).

There is a large amount of literature relating to the determination of pore size distributions and internal surface area for different structures. I find that the authors must do a more convincing job regarding SAXS analysis and include references to literature.

E,g, the authors now write: “On the other hand, more than two peaks, but not exactly indexed as (100), (110), and (200) planes of ordered hexagonal (p6mm) mesostructure, are observed in case of PMSB samples (Fig. 4c-A-C), although other PMSB samples (Fig. 4c-D-F), do not show much significant ordered porosity.” What is meant by ‘not exactly indexed as’?

 Also, I find the sentenceThe values were calculated with the first peak for each sample, however the overall first peaks are very small, representing that the values are not the general largest pore-to-pore distance but the trace attributed from partial and small portion of lager pores. Also, multiple SAXS peaks strongly suggest bimodal or overlapped multimodal nanoporous structure” to be a wrongful interpretation of the SAXS curves.

Overall, I do not find the SAXS analysis convincing, although the general conclusions regarding differences in degree of porosity might hold.

 In the abstract there are still typos (small angel X-ray…) and use of a phrase (magnetic resonance and so on) which is not appropriate language in the context. The authors should not expect referees to find and correct all  language errors.
